# Laboratory and UAV-Based Identification and Classification of Tomato Yellow Leaf Curl, Bacterial Spot, and Target Spot Diseases in Tomato Utilizing Hyperspectral Imaging and Machine Learning

**Jaafar Abdulridha [1], Yiannis Ampatzidis [1,\*], Jawwad Qureshi [2] and Pamela Roberts [3]**

[1] Department of Agricultural and Biological Engineering, Southwest Florida Research and Education Center, University of Florida, 2685 FL-29, Immokalee, FL 34142, USA; ftash@ufl.edu

[2] Department of Entomology and Nematology, Southwest Florida Research and Education Center, University of Florida, 2685 FL-29, Immokalee, FL 34142, USA; jawwadq@ufl.edu

[3] Department of Plant Pathology, Southwest Florida Research and Education Center, University of Florida, 2685 FL-29, Immokalee, FL 34142, USA; pdr@ufl.edu

\* Correspondence: i.ampatzidis@ufl.edu

**Abstract:** Tomato crops are susceptible to multiple diseases, several of which may be present during the same season. Therefore, rapid disease identification could enhance crop management consequently increasing the yield. In this study, nondestructive methods were developed to detect diseases that affect tomato crops, such as bacterial spot (BS), target spot (TS), and tomato yellow leaf curl (TYLC) for two varieties of tomato (susceptible and tolerant to TYLC only) by using hyperspectral sensing in two conditions: a) laboratory (benchtop scanning), and b) in field using an unmanned aerial vehicle (UAV-based). The stepwise discriminant analysis (STDA) and the radial basis function were applied to classify the infected plants and distinguish them from noninfected or healthy (H) plants. Multiple vegetation indices (VIs) and the M statistic method were utilized to distinguish and classify the diseased plants. In general, the classification results between healthy and diseased plants were highly accurate for all diseases; for instance, when comparing H vs. BS, TS, and TYLC in the asymptomatic stage and laboratory conditions, the classification rates were 94%, 95%, and 100%, respectively. Similarly, in the symptomatic stage, the classification rates between healthy and infected plants were 98% for BS, and 99–100% for TS and TYLC diseases. The classification results in the field conditions also showed high values of 98%, 96%, and 100%, for BS, TS, and TYLC, respectively. The VIs that could best identify these diseases were the renormalized difference vegetation index (RDVI), and the modified triangular vegetation index 1 (MTVI 1) in both laboratory and field. The results were promising and suggest the possibility to identify these diseases using remote sensing.

**Keywords:** hyperspectral; artificial intelligence; spectral analysis; UAV; disease detection; classification; machine learning

---

## 1. Introduction

Disease identification can be a very complicated procedure because it needs experienced personnel and frequent field monitoring. Any delay or misidentification of a disease might increase the spread within the field and an economic loss; several diseases need only few days to spread rapidly in the field. Currently, growers use mainly visual observations to identify a disease in the field, targeting early disease detection, in order to apply the right treatment(s) and control the spread. In the last few decades, several technologies have been developed to help the identification of economically

important plant diseases in the laboratory and under field conditions by using nondestructive and remote sensing methods [1–3]. For example, multispectral and hyperspectral sensing has been utilized for pest, plant disease, and stress detection with promising results [4–6]. These sensors measure the light reflection from an object (e.g., plant canopy). Any change (e.g., caused by a disease) that might occur and disturb the canopy or the leaf surface would affect the light reflectance and diffuse the light direction. By detecting these changes, it is possible to identify abnormalities in plants. The most important benefit of these remote sensing technologies and techniques is that they could detect and distinguish a disease even in an asymptomatic stage (before symptoms become obvious to direct visual observations), and hence, this early detection could help to efficiently control and manage a disease and its spread. Martinelli et al. [7] explained that traditional methods, such as DNA-based and serological methods, are still important tools to precisely diagnose plant diseases, but those methods are not designed to detect a disease in an asymptomatic stage. On the other hand, several studies demonstrated the capabilities of several remote sensing techniques for detecting a disease even in an early disease development stage. Whetton et al. [8] utilized hyperspectral imaging to detect yellow rust and Fusarium head blight in cereal crops and determined the best wavelengths (500–700 nm) that were capable to distinguish the diseases in early development stages.

Unmanned aerial vehicles (UAVs) are becoming increasingly available and provide a low-cost solution for field surveying and monitoring, as they can cover large areas in a short time and at relatively low cost. UAVs are now widely used in agriculture for various purposes, including plant disease detection, yield prediction, tree inventory development, plant variety evaluation [9–11]. Jimenez-Brenes et al. [12] developed a UAV-based multispectral imaging technique to detect two weed species in grapevines and create a management map based on the weed distribution. Albetis et al. [12] utilized UAV multispectral sensing to distinguish between two vineyard diseases (Flavescence dorée and Grapevine Trunk) that produce similar symptoms in most of the red grape varieties. Abdulridha et al. [13] developed a UAV-based technique to detect citrus canker in the field by utilizing hyperspectral imaging and machine learning. This technique was able to detect the disease even in the early development stages. Zhang et al. [14] developed an automated technique to detect yellow rust in winter wheat (in five disease development stages) using a deep convolutional neural network (DCNN) to analyze and classify high spatial resolution spectral data acquired by a UAV-based hyperspectral imaging system.

Recognizing the plant stress levels could be the first step for precision management, but it is also important to early identify what caused the stress. For example, early identification of a specific disease in a crop could help the grower/manager to make the right decisions by selecting the best chemical application or employing other appropriate management practices. Several studies utilized vegetation indices (VIs) to identify and distinguish among diseases and other disorders. Since the spectral reflectance of vegetation in the visible region of the electromagnetic range is mainly affected by the chlorophyll pigment concentration [15–17], several projects have studied and analyzed this relationship (between pigment concentration and optical properties of leaves) by utilizing VIs [18–20]. Furthermore, the relationship between canopy spectral reflectance and plant canopy structure was studied in order to develop techniques for disease and stress detection [21,22]. The wavelengths located in the visible (VIS) and near infrared (NIR) range have been proven to be critical for identifying abnormalities in vegetation [23–25].

Multiple diseases attack tomato crops in Florida at the same time, with most of them able to cause significant economic loss. For example, the tomato yellow leaf curl (TYLC), which is transmitted by adult Silverleaf whitefly Bemisia tabaci Biotype B causes serious damage. Current management of this vector disease complex relies on chemical control of whitefly and use of tomato TYLC-tolerant cultivars. If infected early, plant growth is stunted, and plants are hardly able to produce fruit. Other symptoms of this disease include yellow leaf edges, upward leaf cupping, leaf mottling, small leaves, and fruit drop. However, identification based on symptoms alone is not enough, because similar symptoms may result from other viruses or growth conditions. Two other devastating diseases

in Florida are the bacterial spot (BS) and the fungal diseases target spot (TS), which cause fruit spots, making the fruit unmarketable. The symptoms of BS begin as small, yellow-green lesions on young leaves or as dark, water-soaked, greasy-appearing lesions on older foliage that become tan to brownish-red. BS lesion shape is defined by leaf veinlets, so the shape is angular rather than round, which is more typical of fungal leaf spots or toxic symptoms from a pesticide or other chemical spray. The symptoms of TS are very similar to the symptoms of BS disease; they begin as small dark lesions that enlarge to form light brown lesions with a concentric pattern and yellow halo. Therefore, rapid and early disease detection and classification methods are needed not only to detect the disease in early stages, but to implement effective management practices to reduce severity and spread. Identifying a disease in early disease development stages and distinguishing it from other factors that cause similar symptoms in a tomato crop is a very challenging task. Few studies have utilized spectral reflectance techniques to detect tomato diseases. For example, Lu et al. [26] detected multiple tomato diseases (late blight, target, and bacterial spot) in laboratory conditions, at different disease development stages (asymptomatic, early stage, and late stage) using spectroradiometers. Abdulridha et al. [4] detected TS and BS using hyperspectral imaging and VIs in laboratory and field conditions in different disease development stages.

Most of the previous studies developed remote sensing techniques to detect tomato diseases in laboratory conditions. This study develops novel techniques, utilizing hyperspectral imaging and machine learning, to detect TYLC-infected tomato plants in susceptible and tolerant varieties, and distinguish them from healthy plants, and plants infected by two other critical tomato diseases, TS and BS. A classification method and several VIs were used to distinguish between healthy and infected plants in asymptomatic and symptomatic stages. This study developed novel techniques that could be used in laboratory and field (UAV-based) conditions.

## 2. Materials and Methods

For each experiment, plants were physically separated in the field and plants were inoculated with either TS or BS. Plants were naturally infected with TYLC and a plant pathologist confirmed the pathogen associated with the foliar symptoms as either bacterial spot or the fungal pathogen of target spot. The healthy plants were not inoculated and grown in a separate field away from the infected plants, and it was confirmed by the experts that they were not infected with any disease. Similarly, separate fields were selected for each case study (TYLC-, BS-, and TS-infected plants).

### 2.1. Tomato Yellow Leaf Curl (TYLC) Sample Collection

Tomato leaves exhibiting symptoms of TYLC were collected from an experimental field at the Southwest Florida Research and Education Center (SWFREC) in Immokalee, USA. Tomato seedlings of TYLC-tolerant "Charger" or susceptible "FL-47" cultivars were planted in spring 2019. Forty leaves were collected each from tolerant and susceptible varieties in asymptomatic and symptomatic stage. Leaves from the tolerant variety were without any visible symptoms (Figure 1a), while leaves collected from the susceptible variety showed curling, severe stunting, reduced leaf size, and chlorosis (Figure 1b). The UAV data collection and laboratory leaves collection were done after 50–60 days after transplanting on 30 April 2019 at 12:00–2:00 p.m. (Figure 2).

### 2.2. Target Spot and BS Sample Collection

Field preparation and collection of leaves infected with TS and BS were described in a previous study [4] (Figure 1c,d). These experiments were conducted at the SWFREC. Tomato seedling of TS and BS diseases "FL-47" cultivars were planted in spring 2019. Guidelines established by the University of Florida/IFAS were followed for land preparation, fertility, irrigation, weed management, and insect control. Beds were 0.81 m wide with 1.83 m centers covered with black polyethylene film. Each plot consisted of ten plants spaced 0.46 m apart within a 4.57 m row with 3.05 m between each plot. The leaf collection and UAV data collection were on the same day for BS and TS, 6 November 2018.

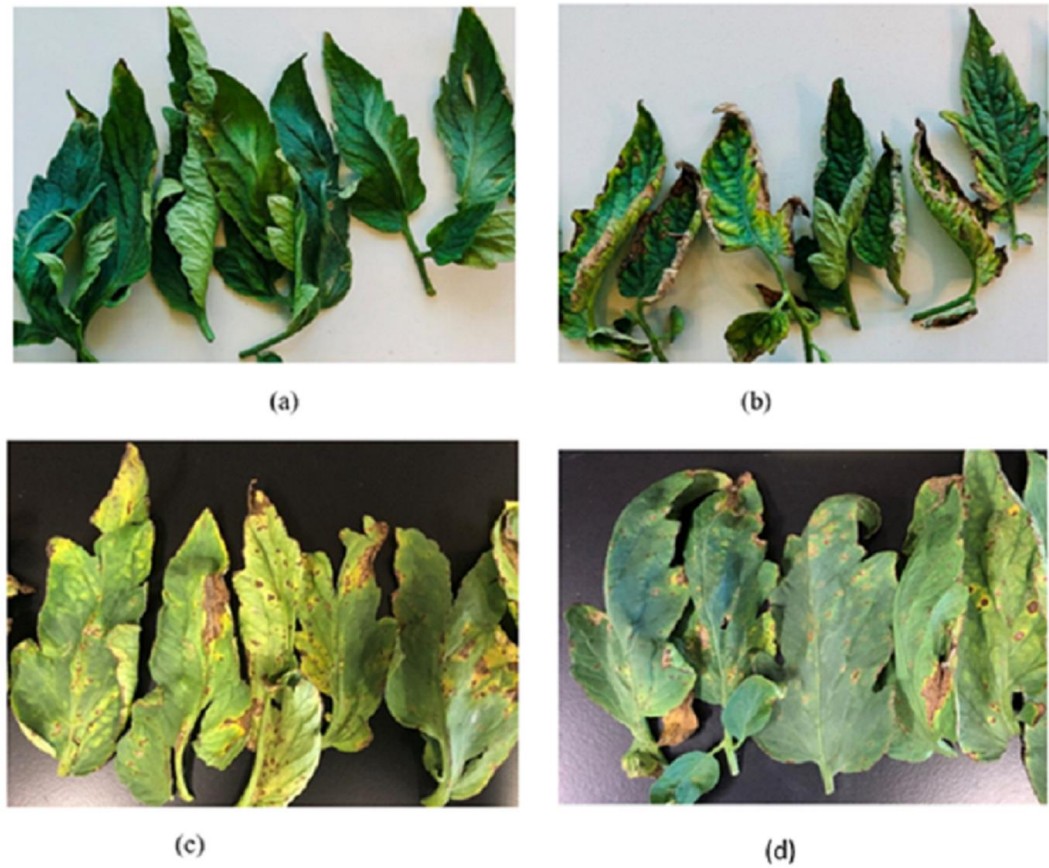

**Figure 1.** Tomato leaves collected to measure their spectral reflectance—tomato yellow leaf curl (TYLC)-infected leaves: (**a**) tolerant variety, (**b**) susceptible variety, and leaves infected by (**c**) target spot and (**d**) bacterial spot.

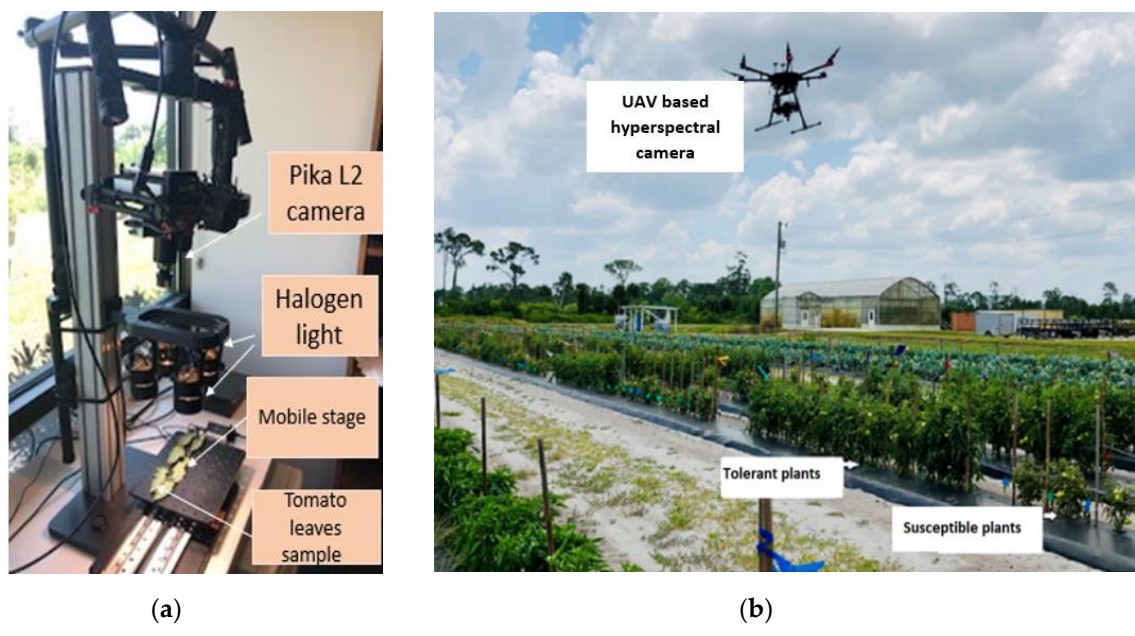

**Figure 2.** (**a**) Benchtop Pika L 2.4 camera with leaf samples in the laboratory condition. (**b**) Unmanned aerial vehicle (UAV) using hyperspectral sensing to collect spectral reflectance data from susceptible and tolerant tomato plants infected with tomato yellow leaf curl. The data were collected after 50–60 days from transplanting on 30 April 2019.

### 2.2.1. Inoculation Methods

Tomato Yellow Leaf Curl Disease

Naturally occurring populations of the viruliferous whitefly vector were sufficient to infect plants which was apparent from the symptoms on the plants.

Target Spot

Plants in plots were inoculated with *Corynespora cassiicola* on 15 October 2018. Cultures of CC #19 and CC #20 (kindly provided by Gary Vallad) were grown for 14 days on one-quarter potato dextrose agar + rifampicin and ampicillin (MilliporeSigma, St. Louis, MO, USA). Plates were flooded with sterile water and fungi was scraped from the surface. The suspension containing mycelium and spores was filtered through three layers of cheese cloth and adjusted to approximately $10^4$ spores mL$^{-1}$ in sterile water. Inoculum was applied with a hand pump sprayer and plants were sprayed to runoff. Confirmation of TS lesions was through microscopic examination of the lesions for presence of spores. No other disease was identified on these plants.

Bacterial Spot

Plants were inoculated with *Xanthomonas perforans* races 3 and 4 on 17 October 2018. Bacteria were grown in 25 mL of Difco Nutrient Broth (NB) overnight on a shaker incubator and transferred to 500 mL NB and incubated as before for 24 h. Bacterial suspension was adjusted to $10^6$ CFU mL$^{-1}$ and applied to tomato plants to runoff using a hand pump sprayer. Lesions of bacterial spot were confirmed by re-isolation of the bacteria onto NA.

### 2.3. Laboratory Data Collection

Four leaves were collected from ten plants for each individual disease stage, i.e., a total of 240 leaves for two stages each for all three diseases, and the spectral reflectance was measured for each leaf individually. Each leaf was scanned by a benchtop hyperspectral imaging system, Pika L 2.4 (Resonon Inc., Bozeman, MT, USA). This sensing system is equipped with a 23 mm lens that has a wavelength range of 380–1020 nm, 281 spectral channels, 15.3° field of view, and a spectral resolution of 2.1 nm. This system falls into the category of line-scan imagers, also known as push broom imagers. The scanner stage is powered by a motor for its linear movement. Four halogen light sources were mounted up at 50 cm above the scanning stage to provide ideal lighting conditions for carrying out the scans (Figure 2a). A white tile was used for calibration of the spectral data, which was provided by the manufacturer of the hyperspectral camera. Leaves were collected in the field by specialists who determined the disease development stages. These leaves were kept in a cooler and transferred to the laboratory immediately, so there was no delay that might affect the quality of the leaf samples. Four leaves were placed on the stage and the leaves were scanned similarly. For each leaf, six regions of interest (RoI) were selected from which the spectral reflectance was measured and recorded. The various RoI included spots with and without symptoms. A total of forty leaves (for both asymptomatic and symptomatic stages) were selected for each disease development stage, and hence, the total spectral reflectance samples for each case study was 40 leaves. The leaves were collected after 50–60 days from transplanting. All data were saved in a text file and then exported to an Excel spreadsheet using postprocessing data analysis software (Spectronon Pro, Resonon Inc., Bozeman, MT, USA).

### 2.4. UAV-Based Data Collection

Spectral data were collected using a UAV (Matrice 600 Pro Hexacopter, DJI, Shenzhen, China) and the same Resonon Pika L 2.4 hyperspectral imaging system (Figure 2), as used in the laboratory conditions. The UAV-based imaging system included: (i) Pika L 2.4 hyperspectral camera (Resonon,

Bozeman, MT, USA); (ii) visible–near infrared (V–NIR) objective lenses for the Pika L camera, with a focal length of 17 mm, a field of view (FOV) of 17.6 degrees, and instantaneous field of view (IFOV) of 0.71 mrad; (iii) a global positioning system (GPS) (Tallysman 33-2710NM-00-3000, Tallysman Wireless Inc., Ottawa, ON, Canada)/inertial measurement unit (IMU) (Ellipse N, SGB Systems S.A.S., France) and global navigation satellite system (GNSS) (Tallysman 33-2710NM-00-3000, Tallysman Wireless Inc., Ottawa, ON, Canada)/inertial measurement unit (IMU) (Ellipse N, SGB Systems S.A.S., France) flight control system for the multirotor aircraft's record sensor position and orientation; and (iv) hyperspectral data analysis software (Spectronon Pro, Resonon, Bozeman, MT, USA) with an ability to correct the GPS/IMU data using a georectification plugin. Data were collected at 30 m above the ground at a speed of 1.5 m/s. Regions of interest were selected arbitrarily, 3–4 pixels for each RoI, to avoid background interactions, and the resolution was 1.03 cm per pixel. Six RoI for each plant were selected; forty tomato plants per disease were chosen in this study (Figure 2b).

After the spectral data were acquired, the maps and images were analyzed by using the Spectronon Pro software. The spatial resolution of the spectral data collected was 0.1 m. The RoI were selected manually (arbitrary) for each plant, and spectral scans were performed to ensure that the entire canopy was covered spectrally. The spectral data from each RoI were then exported as a text file and processed using the SPSS software (SPSS 13.0, Inc., Chicago; Microsoft Corp., Redmond, WA, USA).

*2.5. Classification Methods*

Two classification methods were utilized in this study. The first method was the stepwise discriminate method (STDA), which is commonly used in agricultural research to select valuable subjects of variables and to estimate the order of the importance of each variable [27]. For example, the STDA method was applied in remote sensing applications to identify patterns in datasets and to determine the probability of a dataset (spectral data) belonging to a given group. This method utilizes backward elimination to remove features/factors that do not have a significant effect on the prediction, while building a machine learning model. Several parameters can be used to determine if a feature/factor significantly affects the prediction, including the Wilks lambda, the Mahalanobis distance, and the F value. If the F value of a variable is statistically significant in the discrimination group, then it means that the variable contributes to the estimation of group participation [28]. In this study, the input data contained the spectral reflectance of three tomato diseases (TYLC-tolerant and susceptible varieties, TS, and BS). The training dataset was 70%, and the testing dataset was 30%.

The second classification method was the radial basis function (RBF). Radial basis function is an artificial neural network that uses radial basis functions as activation functions. The output of the network is a linear combination of radial basis functions of the inputs and neuron parameters. Once the files were prepared in SPSS format, 70% training and 30% test were obtained for each disease versus healthy leaves. We also tried to classify between the diseases that had the same symptoms in early or late stages, which classified between TYLC-tolerant and susceptible varieties, and also to classify between BS and TS diseases in two stages. Forty leaves were the sample size for each disease and stage. Bacterial spot and target spot were analyzed individually without tolerant and susceptible varieties. The same process was used for the field condition, except there was no asymptomatic stage; this was because we were unable to identify which canopy had a higher rate of disease infection. The classification of field condition was between healthy vs. TYLC-tolerant and susceptible, healthy vs. BS, and H vs. TS disease. The training set was used to train discrimination for classifying tomato diseases and the test dataset was used to evaluate the classification accuracy for the different models.

*2.6. Vegetation Indices*

As the light reaches a surface, part of it is reflected, part of it is transmitted, and the rest is absorbed. The amount of light that is reflected, transmitted, and absorbed depends upon the characteristics of the surface, and it changes with the wavelength of the light. For example, most of the light reaching the soil is either reflected or absorbed while the amount of light being transmitted is very low. The change

with wavelengths is also relatively low. However, this is not the case with vegetation. In near-infrared wavelengths, the most light is transmitted and reflected while the amount of light being absorbed is very low; in the visible range, however, it is the opposite.

The main goal of the majority of remote sensing studies is to detect and identify the condition of vegetation in a specific area(s). The amount and composition of solar irradiance that reaches the surface determine the amount of light reflected from it, thus the reflectance properties of the surface. As solar irradiance varies with the period of day and weather, a simple measurement of light reflected from a surface may not be an accurate way to characterize the surface in a repeatable manner. This difficulty can be overcome by using relevant measurements, for example, combinations of reflectance data from two or more spectral bands, to form what is known as a vegetation indices (VIs). The use of spectral band data can be in different forms such as ratioing, differencing, ratioing differences, and sums, and by forming linear combinations. Vegetation indices are designed in a way such that they present effective vegetation characteristics by reducing solar irradiance and soil background effects. Most VIs are designed so they can be used in a consistent manner and be applicable and accurate over time and in different locations. Their values can be affected by many factors, such as viewing angles, solar irradiance, intervening atmosphere (among others), which is especially true in cases of data obtained from aerial platforms. Vegetation index values can vary even if they are collected over the same area, but with different instruments due to varying specifications of the instruments (and calibration procedures). In this study, several VIs were selected (Table 1), based on similar studies, to evaluate their effectiveness on distinguishing among different diseases and disease development stages. Since diseases develop different symptoms and physiological changes in a plant, VIs can be utilized to detect and quantify these morphological and physiological changes for each disease and disease development stage.

**Table 1.** Vegetation indices evaluated for disease detection.

| Vegetation Indices | Equations | References |
|---|---|---|
| Ratio Analysis of reflectance Spectral Chlorophyll-a (RARSa) | $RARSa = \frac{R675\ nm}{R700\ nm}$ | [29] |
| Ratio Analysis of reflectance Spectral Chlorophyll b (RARSb) | $RARSb = \frac{R675\ nm}{(R700\ nm \times R650\ nm)}$ | [29] |
| Ratio analysis of reflectance spectra (RARSc) | $RARSc = \frac{R760\ nm}{R500\ nm}$ | [29] |
| Pigment specific simple ratio (PSSRa) | $PSSRa = \frac{NIR800\ nm}{R680\ nm}$ | [30] |
| Normalized difference vegetation index 780 (NDVI 780) | $NDVI\ 780 = \frac{R780\ nm - R670\ nm}{R780\ nm + R670\ nm}$ | [31] |
| Structure Insensitive Pigment Index (SIPI) | $SIPI = \frac{(NIR840\ nm - R450\ nm)}{(NIR840\ nm - R670\ nm)}$ | [32] |
| Normalized phaeophytinization index (NPQI) | $NPQI = \frac{(R415\ nm - R435\ nm)}{(R415\ nm + R435\ nm)}$ | [33] |
| Red-Edge Vegetation Stress Index 1 (RVS1) | $RVSI1 = \left[\frac{(R651\ nm + Red\ Edge\ 750\ nm)}{2}\right] - Red\ Edge\ 733\ nm$ | [34] |
| Triangle Vegetation Index (TVI) | $TVI = 0.5[120 * (R761\ nm - R581\ nm) - 200(R651\ nm - R581\ nm)]$ | [35] |
| Renormalized Difference Vegetation Index (RDVI) | $RDVI = \frac{(R761\ nm - R651\ nm)}{\sqrt{(R761\ nm + R651\ nm)}}$ | [36] |
| Normalized difference vegetation index 850 (NDVI850) | $NDVI850 = \frac{(NIR850\ nm - R651\ nm)}{(NIR850\ nm + R651\ nm)}$ | [31] |
| Simple Ratio Index (SR 761) | $SR\ 761 = \frac{R761\ nm}{R651\ nm}$ | [37] |
| Simple Ratio Index (SR 850) | $SR\ 850\ nm = \frac{NIR850\ nm}{R650\ nm}$ | This study |
| Simple Ratio Index (SR 900) | $SR\ 900\ nm = \frac{NIR900\ nm}{R680\ nm}$ | This study |
| Water Stress and Canopy Temperature (NWI 2) | $NWI\ 2 = \frac{NIR970\ nm - NIR850\ nm}{NIR970\ nm + NIR850\ nm}$ | [38] |
| Green NDVI (GNDVI) | $GNDVI = \frac{(NIR850\ nm - R580\ nm)}{(NIR850\ nm + R580\ nm)}$ | [39] |

**Table 1.** *Cont.*

| Vegetation Indices | Equations | References |
|---|---|---|
| Photochemical Reflectance Index (PRI) | $\textbf{PRI} = \frac{(\textbf{R531 nm} - \textbf{R570 nm})}{(\textbf{R531 nm} + \textbf{R570 nm})}$ | [40] |
| Modified Chlorophyll Absorption in Reflectance Index (mCARI 1) | $\textbf{mCARI 1} =$ $\mathbf{1.2}\big[(\mathbf{2.5} * \textbf{R761 nm} - \textbf{R651 nm}) - \mathbf{1.3}(\textbf{R761 nm} - \textbf{R581 nm})\big]$ | [41] |
| Modified Triangular Vegetation Index1 (MTVI 1) | $\textbf{MTVI 1} =$ $\mathbf{1.2}\big[\mathbf{1.2}(\mathbf{1.2}(\textbf{R760 nm} - \textbf{R580 nm}) - \mathbf{2.5}(\textbf{R650 nm} - \textbf{R580 nm})\big]$ | [41] |
| Modified Triangular Vegetation Index2 (MTVI 2) | $\textbf{MTVI2} = \frac{\mathbf{1.5}[\mathbf{1.2}(\textbf{R760 nm} - \textbf{R580 nm}) - \mathbf{2.5}(\textbf{R650 nm} - \textbf{R580 nm})]}{\sqrt{[(\mathbf{2}*\textbf{R760 nm} + 1)2 - (\mathbf{6}*\textbf{R760 nm} - \mathbf{5}* \sqrt{(\textbf{R650 nm})} - 0.5}}$ | [41] |

Data Analysis for Selecting VIs

The M statistic value (Equation (1)) was used to evaluate the effectiveness of each VI to distinguish tomato plants infected by TS, TB, and TYLC in different disease development stages. Generally, the M statistic value of a VI is higher when the standard deviation is low, which helps to better identify the best VI for disease detection [42]. When it is more than one, the M statistic value is considered to be a significant discriminant between different VIs; when the M statistic value increases, it means more and better overlap separation.

$$statistic value = \frac{\left(Mean_{Healthy} - Mean_{Infected}\right)}{\left(\sigma_{Healthy} + \sigma_{Infected}\right)} \tag{1}$$

## 3. Results and Discussion

### 3.1. Spectral Reflectance Analysis

In this section, a detailed analysis of the spectral reflectance of TYLC-infected tomato plants, in a tolerant and susceptible varieties, is presented and compared with the spectral reflectance of healthy plants and plants infected by TS and BS. A detailed spectral analysis of tomato plants infected by TS and BS is presented by [4].

3.1.1. Spectral Reflectance of TYLC, BS, and TS Diseases: Laboratory-Based Analysis

Since we were examining plants exposed to natural inoculation via the whitefly vector, we could not determine precisely when individual plants were inoculated. However, as the season progressed, the severity of symptoms also progressed throughout the season. Therefore, we will refer to "early stages" or "asymptomatic" as the plant was asymptomatic (no visual symptoms) and "advanced stages" or "symptomatic" as visual symptoms present later in the season. Leaves examined early in the season did not show visual symptoms. However, there were significant differences when comparing the spectral reflectance signatures of infected plants in different disease development stages (Figure 3a). The spectral values of both tomato varieties (tolerant and susceptible to TYLC) in the asymptomatic stage were lower than the spectral values of the healthy plants in the green range. The spectral signature of BS and TS also showed different spectral reflectance in the red and green ranges; the reflectance of spectral was increased or decreased based on disease severity. The red range showed higher reflectance for BS and TS than for the healthy plants. In NIR, the BS showed a lower reflectance value for asymptomatic stage [4].

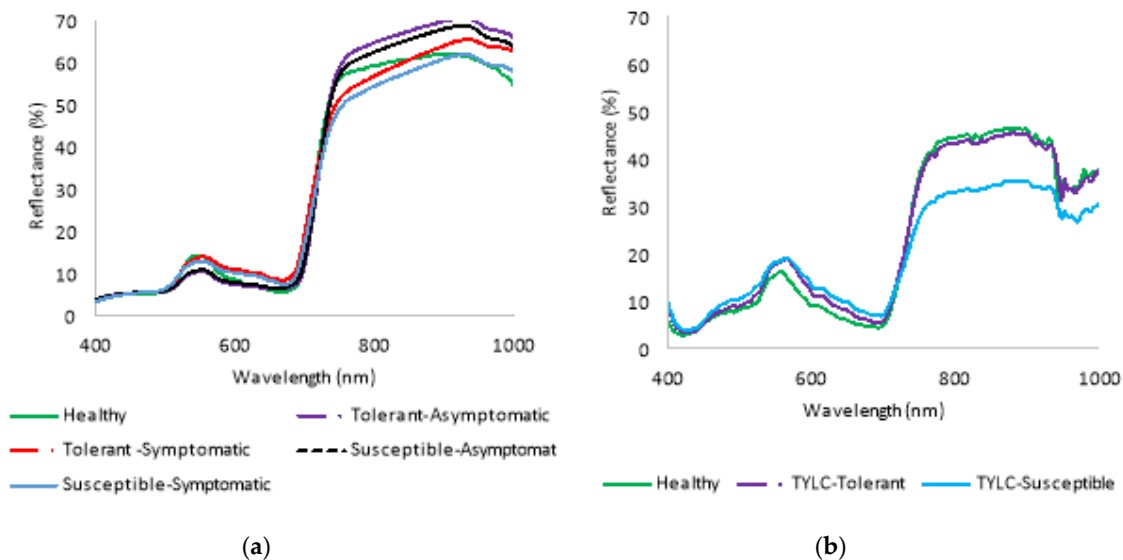

**Figure 3.** The spectral reflectance signatures of healthy, TYLC-infected (tolerant and susceptible varieties) tomato plants: (**a**) in laboratory conditions and (**b**) field conditions.

### 3.1.2. Spectral Reflectance of TYLC, BS, and TS Diseases: Field (UAV)-Based Analysis

In the field, it was difficult to determine the disease development stages, because in most cases plants included leaves with early symptoms (early disease development stages), and late symptoms (late stages) as well. Figure 3b presents the spectral signatures of the healthy (H) and TYLC-infected (tolerant and susceptible varieties) tomato plants, generated by the UAV-based hyperspectral sensing system. No significant differences were noticed between H and TYLC-tolerant plants in the NIR range; however, significant differences occurred in the red range (Figure 3b). In the susceptible variety, the spectral reflectance signature shifted down in NIR (similar to the laboratory conditions). In the field, the spectral signature of BS and TS recorded the lowest values in the visible range, compared to TYLC-infected and healthy plants, while the reflectance values increased in NIR range (Figure 4b). The spectral signature of the TYLC-infected susceptible variety was significantly different, especially in the NIR range, from the BS and TS signatures.

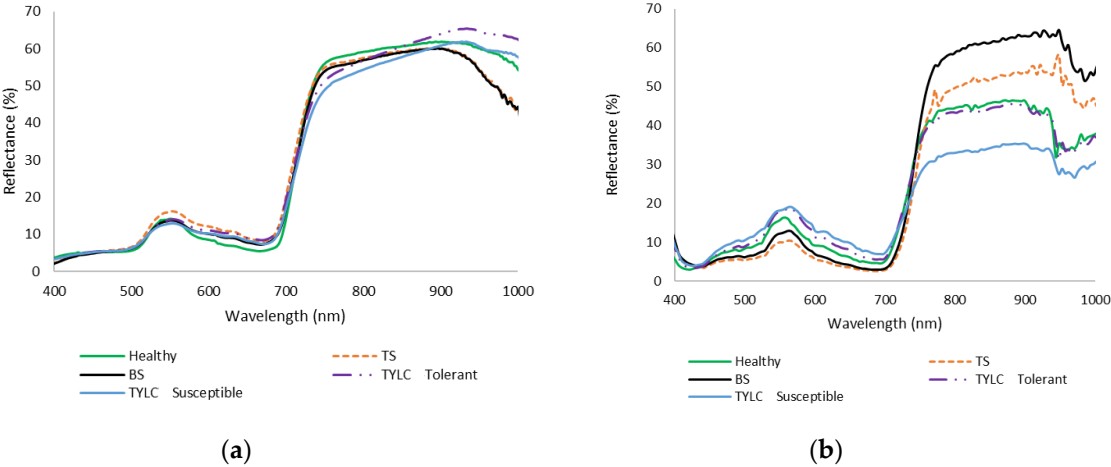

**Figure 4.** The spectral reflectance signatures of TYLC (on susceptible and tolerant tomato varieties), bacterial spot (BS), and target spot (TS) infected tomato plants: (**a**) in laboratory conditions and (**b**) field conditions (UAV-based).

### 3.2. Classification Results

In laboratory conditions, the classification results between H and TYLC-infected plants (tolerant and susceptible varieties) were 100% for both disease development stages (asymptomatic stage and symptomatic stage) (Table 2). The classification results between H and BS or TS were also high, reaching up to 95% in STDA, while recording a lower rate in RBF, 82–83%. The RBF method was less accurate than the STDA classification method in the major classification categories. In the field conditions, similar classification results were recorded (Table 2) between H and all diseases. Wilks lambda recorded a very low value, indicating strong separation value between H and other categories. The scale ranges from 0 to 1, where 0 means total discrimination, and 1 means no discrimination. The lower Wilks lambda value (0.014) represents maximum discrimination among other categories. A higher value of chi-square means a significant difference between the variables. The cross validation and overall percent were identical for most of the category classes. The classification rate between TYLC tolerant vs. TYLC susceptible in the asymptomatic stage was 100% for both STDA and RBF methods. The classification of TYLC-tolerant vs. TYLC-susceptible varieties in the symptomatic stage was 100% in STDA, while the RBF method showed a lower classification value of 66.7%. The STDA classification in the laboratory and field produced higher classification values than the RBF method for most of the categories (Table 2).

**Table 2.** Classification results utilizing the stepwise discriminant analysis (STDA) and the radial basis function (RBF) for TYLC-, BS-, and TS-infected tomato plants in symptomatic (Sym) and asymptomatic (Asy) stages.

| Parameter | STDA | | | | RBF (%) |
|---|---|---|---|---|---|
| | Overall Percent (%) | Cross Validation (%) | Wilks Lambda | Chi-Square | |
| Laboratory based | | | | | |
| H vs. TYLC Tolerant-Asy | 100 | 100 | 0.014 | 388.0 | 89 |
| H vs. TYLC Tolerant-Sym | 100 | 100 | 0.028 | 374.1 | 100 |
| H vs. TYLC Susceptible-Asy | 100 | 100 | 0.023 | 320.2 | 83 |
| H vs. TYLC Susceptible -Sym | 100 | 100 | 0.044 | 321.4 | 100 |
| Tolerant-Asy vs. Susceptible-Asy | 100 | 100 | 0.086 | 91.8 | 100 |
| Tolerant-Sym vs. Susceptible-Sym | 100 | 100 | 0.096 | 111.1 | 66.7 |
| H vs. TS-Asy | 95 | 95 | 0.045 | 523.6 | 82 |
| H vs. TS-Sym | 95 | 95 | 0.005 | 422.4 | 90 |
| H vs. BS-Asy | 94 | 94 | 0.005 | 924.1 | 95 |
| H vs. BS-Sym | 95 | 94 | 0.005 | 523.6 | 89 |
| TS-Asy vs. BS-Asy | 88 | 87 | 0.306 | 145.0 | 83 |
| Ts-Sym vs. BS-Sym | 82 | 82 | 0.456 | 120.3 | 46 |
| Field (UAV) based | | | | | |
| H vs. TYLC Tolerant | 100 | 100 | 0.006 | 539.9 | 100 |
| H vs. TYLC Susceptible | 100 | 100 | 0.018 | 378.1 | 97 |
| TYLC Tolerant vs. Susceptible | 100 | 100 | 0.104 | 146.2 | 76 |
| H vs. TS | 98 | 96 | 0.026 | 597.8 | 98 |
| H vs. BS | 96 | 96 | 0.013 | 541.5 | 93 |
| Ts vs. BS | 82 | 80 | 0.457 | 141.9 | 64 |

### 3.3. Significant VIs for Disease Detection: Laboratory-Based Analysis

Most of the VIs' M statistic values for the BS-, TS-, and TYLC-susceptible variety were positive (Figure 5), while most M statistic values of the TYLC-tolerant variety were negative. Only a few VIs of the TYLC-susceptible variety have negative M statistic values (RVS, NWI 2, and RARSa). Based on this observation, it is possible to distinguish between TYLC susceptible and all other diseases. The modified triangular vegetation index 1 (MTVI 1) and the renormalized difference vegetation index (RDVI) can

be used to accurately distinguish among all diseases since their M statistic values are significantly different (Figure 5).

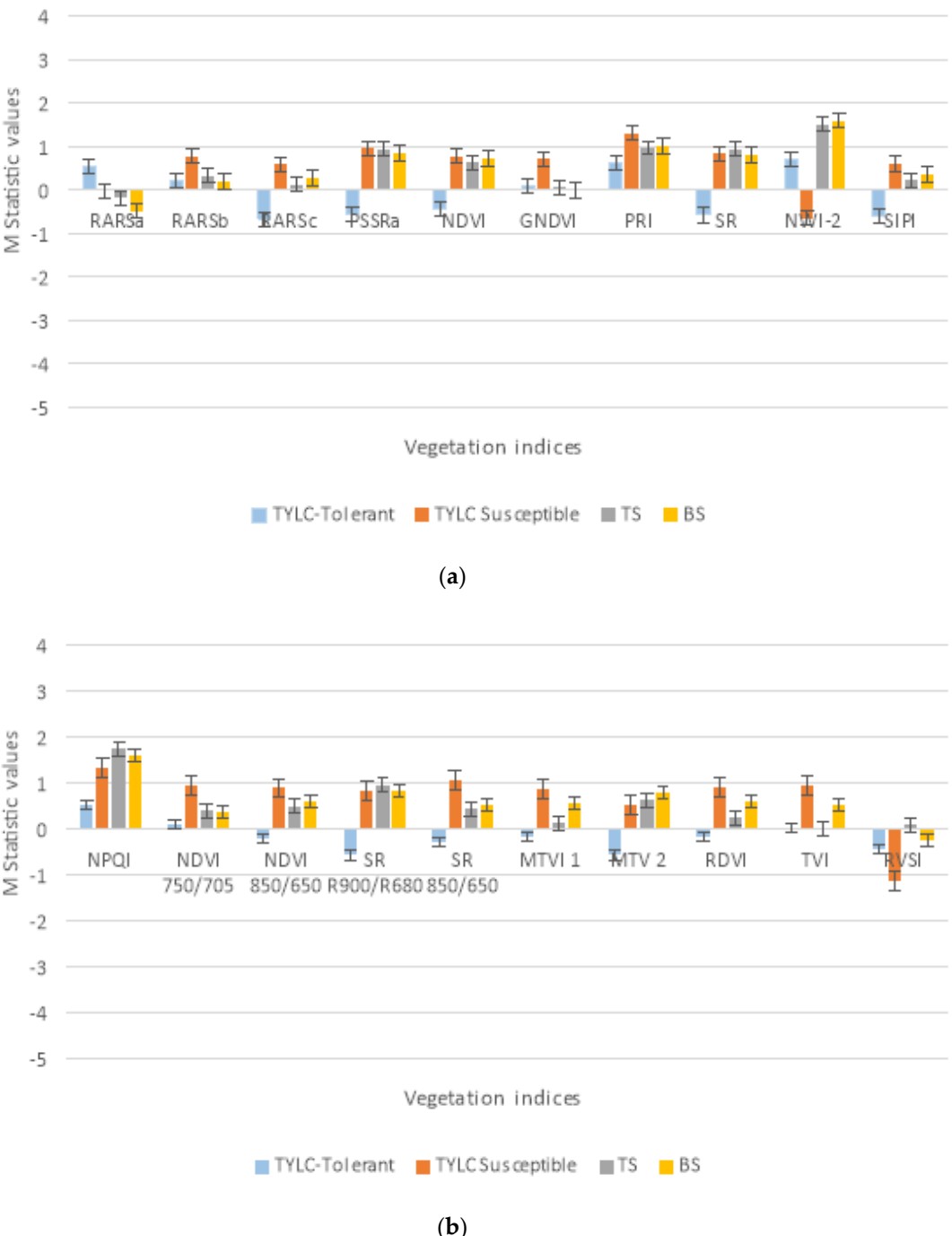

(a)

(b)

**Figure 5.** The M statistic value of vegetation indices for TYLC, BS, and TS diseases in the laboratory: (**a**) first group of VIs and (**b**) second group of VIs. The vertical bar represents the error bar.

## 3.4. Significant VIs for Disease Detection: Field (UAV)-Based Analysis

The same VIs (MTVI 1 and RDVI) can be used to accurately distinguish among all diseases utilizing the UAV-based sensing system and the M statistic value (Figure 6). There are several VIs that can be used to distinguish BS and TS diseases from TYLC (tolerant and susceptible varieties) (Figure 6), but only a few to distinguish among all diseases.

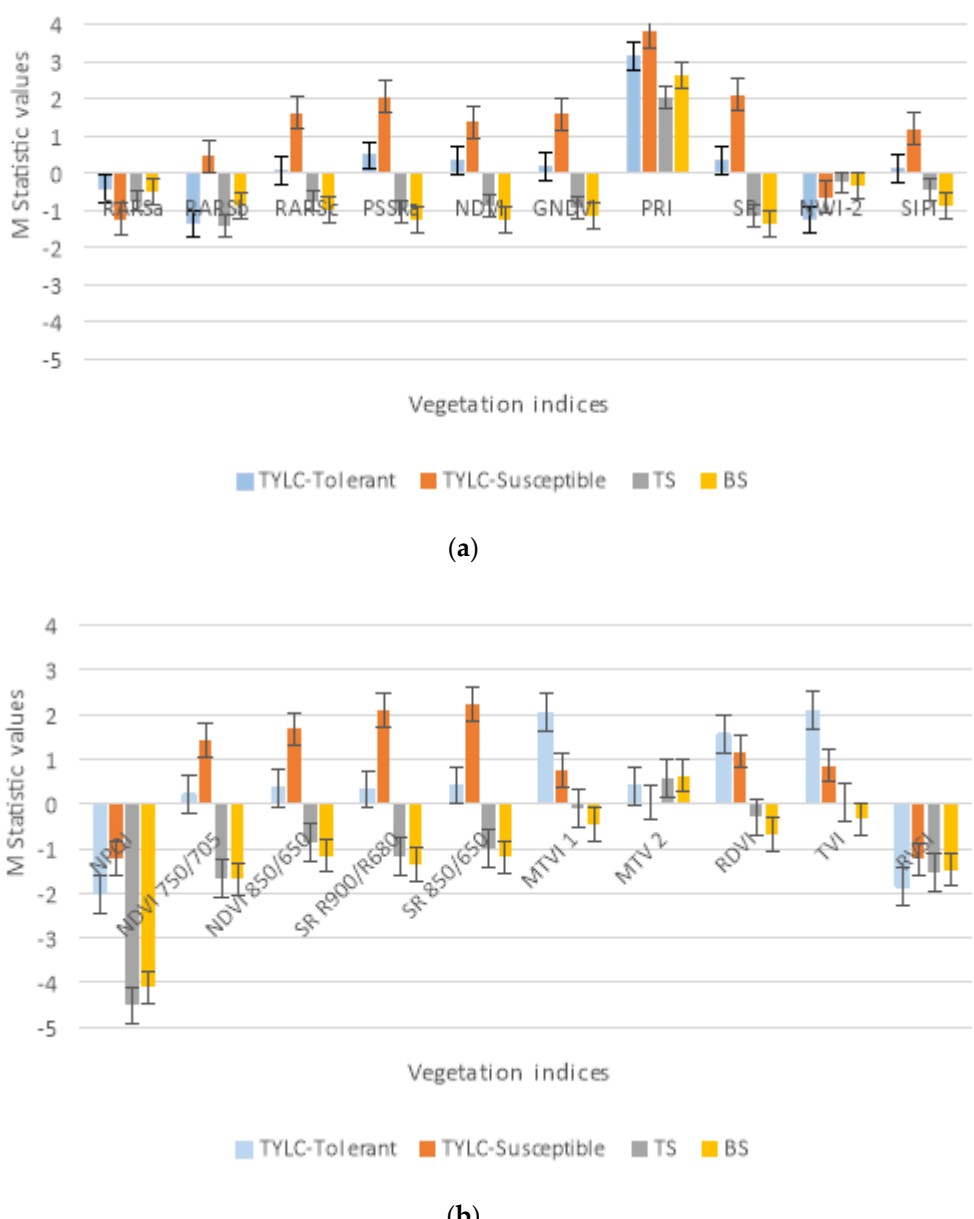

**Figure 6.** The M statistic value of vegetation indices (VIs) for TYLC, BS, and TS diseases in the field (UAV-based): (**a**) first group of VIs, and (**b**) second group of VIs. The vertical lines present error bars.

## 4. Discussion

We assume that in the TYLC-infected plants, the spectral changes in the green range were caused by the deterioration of chlorophyll concentration and were almost impossible to notice with the naked eye. The spectral reflectance in the red edge showed differences between healthy and asymptomatic stages for both tolerant and susceptible varieties. In the symptomatic stage, the spectral reflectance in the red range was higher than the spectral reflectance value of a healthy plant. The spectral reflectance values in the red edge shifted down when compared to healthy plants and the asymptomatic stage. The spectral signature of the asymptomatic stage showed different patterns than the healthy and symptomatic stage in the visible and NIR ranges. It is obvious that the spectral reflectance values were higher in the red range and lower in the red edge. Furthermore, there were several differences in the spectral values for the different disease development stages (disease severity). The leaves of the tolerant variety did not show heavy symptoms in the late disease development stages, except few

necrotic lesions; while the leaves of the susceptible variety showed heavy necrotic lesions, including curl and yellow margin in all leaves.

In general, the spectral reflectance signatures showed different patterns in the TYLC-infected and healthy plants. The spectral reflectance of the diseased plants had increased values in the red range and NIR, indicating a decrease in the chlorophyll concentration, compared to the healthy plants. Reflectance in the red edge (700–800 nm) was changed most greatly with disease development because the pigment concentration was changed, which increased the yellowish color in the leaves. In general, it was possible to identify and distinguish healthy from TYLC-infected plants, as well as the disease development stages, by analyzing and comparing their spectral reflectance signatures; reflectance indices combining reflectance near to 531 nm could give a guide of photosynthetic function [40,43]. The major impact of TYLC susceptibility to leaf reflectance was in the visible range from 550 to 700 nm and in the NIR range from 700 nm to 850 nm, respectively. These wavelength regions are affected by the chlorophyll and brown pigment concentration as well as by the water content and canopy structural changes [44]. Due to plant cell death, membrane damage, and necrotic increase, the virus penetrated the leaf stomata [45]. The variance of the reflectance in the NIR region was mostly affected by situation of leaf tissue or any abnormal process that might affect the inner scatter process. The light division is also influenced by water content and air–water interference [46–49].

It was obvious to notice the plant damage in the late season of TYLC on the susceptible variety. Plants, which were initially stunted, showed severe stunting in the end of the season (Figure 1b). The spectral reflectance showed differences between laboratory and field condition which might refer to the measurement conditions. In the laboratory, the leaf samples were laid down on one flat side and facing up with the halogen light source, with optimal temperature and humidity. Therefore, the reflectance of the light obtained from the leaf samples represent the object in an ideal condition. The spectral reflectance in the field has different reflectance than the laboratory because of different sunlight angle(s) and light density, the field of view angle, and the weather conditions (e.g., clouds). Furthermore, the canopy structure of the plants included different type of leaves of which some were infected and others were not, or had lower infection. In general, there was no proper way to select only symptomatic or asymptomatic leaves from images taken at 30 m height (UAV-based method). For the images acquired by UAV, it was difficult to determine the disease development stages, because in most cases plants included leaves with minor disease development stages and late symptoms.

All diseased plants produced different spectral signatures in the laboratory measurement. The chlorophyll concentration was increased or decreased based on the disease type; this difference can be seen in the visible range and red edge–NIR (690–1000 nm). In the field condition, the spectral signature of the TYLC-infected susceptible variety was significantly different, especially in the NIR, from the BS and TS signatures. These differences in the spectral signatures might be used to distinguish among these diseases. As can be seen from Figure 1a, the TYLC-infected leaves of the tolerant variety in the early disease development stages did not produce any obvious symptoms, and hence, it was very difficult to identify the infection visually. On the other hand, it was possible to detect the TYLC by analyzing and comparing the spectral signatures of all samples. Heim et al. [50] also identified spectral differences in red edge and NIR to detect myrtle rust (Austropuccinia psidii) on lemon myrtle trees, and Romer et al. [51] used the same spectral ranges to detect presymptomatic wheat leaf rust. Each disease showed different symptoms in late disease development stages, while in the early stages or the asymptomatic stage, it was very difficult, even for experts, to diagnose the disease based on symptoms. Many researchers have applied hyperspectral imaging to detect diseases in a very early stage or the presymptomatic stage with promising results [52–55].

## 5. Conclusions

This study developed novel techniques that can be used in the laboratory and field for the identification and classification of three critical tomato diseases, BS, TS, and TYLC (in the tolerant and susceptible varieties), by utilizing hyperspectral imaging and machine learning. In early disease

development stages, similar symptoms were produced by these diseases, and it was very difficult to distinguish among diseases visually. However, utilizing the STDA and RBF classification methods, it was possible to identify and classify the diseased and healthy tomato plants with high accuracies. Furthermore, the M statistic values from several VIs were analyzed, and it was found that the values of two VIs (MTVI 1 and RDVI) were significantly different for all diseases. Hence, these indices can be used to accurately detect and classify these diseases. The M statistic values of the VIs varied between laboratory and field; still, in both conditions, the M statistic values of the MTVI 1 and RDVI showed significant differences among the three diseases.

**Author Contributions:** J.A. collected and analyzed the data. Y.A., J.Q., and P.R. supervised the project and contributed to the research design. All authors contributed to all aspects of the preparation and the writing of the manuscript. All authors have read and agreed to the published version of the manuscript.

**Funding:** This material was made possible, in part, by a Cooperative Agreement from the U.S. Department of Agriculture's Agricultural Marketing Service through grant AM190100XXXXG036. Its contents are solely the responsibility of the authors and do not necessarily represent the official views of the USDA.

**Conflicts of Interest:** This material was made possible, in part, by a Cooperative Agreement from the U.S. Department of Agriculture's Agricultural Marketing Service through grant AM190100XXXXG036. Its contents are solely the responsibility of the authors and do not necessarily represent the official views of the USDA.

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
