# Peer review of "Laboratory and UAV-Based Identification and Classification of Tomato Yellow Leaf Curl, Bacterial Spot, and Target Spot Diseases in Tomato Utilizing Hyperspectral Imaging and Machine Learning"

_remotesensing, doi:10.3390/rs12172732_

Round 1

Reviewer 1 Report

This is an interesting paper related to UAV-based identification and classification of diseases in tomato plants using hyperspectral imaging and machine learning. This is state of the art paper and it is well written and structured. Some specific recommendations are below:

Specific recommendations:

Line 28: add “and” before “99%”. Delete “,” after “TS”.

Keywords: consider to replace all of them with: artificial intelligence, spectral analysis, UAV, disease detection, classification.

Line 68: add space between “[14]” and “developed”.

Line 87: delete “plants” after “and”.

Line 93: Usually when we start a sentence we do not use abbreviation. Consider to replace “BS” with “Bacterial spot”.

Line 97-99: do you repeat this sentence here? I believe you explain that above.

Line 112: do you mean “stages” (not “status”)?

Line 127: why you talk about TYLC in this section?

Line: 129-131: Consider to add the same information for the TYLC (section above).

Line 142: status or stages?

Line 144: add space between “23” and “mm”.

Line 159: Consider to change this sub-title to: “UAV based data collection” to match with the previous sub-title “Laboratory data collection”.

Line 168: please check again this sentence.

Line 190: consider to delete “the” before “specific”.

Line 216: “when the M statistic value increases, it means more and better overlap separation”. Is that correct?

Line 228: consider to add “Foe example” before “If the F…”.

Line 240-244: Does these sentences belong to M&M (maybe the “2.1 Tomato yellow leaf curl (TYLC) sample collection” section)?

Line 290: Do you mean “Figure 4a”?

Line 319: cite the reference here based on journal’s instructions.

Table 2: consider to better describe columns 2 and 4. Maybe “STPD classification results (%)”.

Author Response

Reviewer 1

This is an interesting paper related to UAV-based identification and classification of diseases in tomato plants using hyperspectral imaging and machine learning. This is state of the art paper and it is well written and structured. Some specific recommendations are below:

Thank you for your comments and suggestions.

Specific recommendations:

Line 28: add “and” before “99%”. Delete “,” after “TS”.

Line 28: It was added

Keywords: consider to replace all of them with: artificial intelligence, spectral analysis, UAV, disease detection, classification.

It was corrected as suggested “Hyperspectral, artificial intelligence, spectral analysis, UAV, disease detection, classification, machine learning.”

Line 68: add space between “[14]” and “developed”.

Line 67: It was added

Line 87: delete “plants” after “and”.

It was deleted

Line 93: Usually when we start a sentence we do not use abbreviation. Consider to replace “BS” with “Bacterial spot”.

Line 94: Bacterial spot is not in the beginning of sentence in this case “The symptoms of BS begin as small, yellow-green lesions on young leaves or as dark..”

Line 97-99: do you repeat this sentence here? I believe you explain that above.

We mentioned the similarity of symptoms in some diseases but not BS and TS in previous paragraph.

Line 112: do you mean “stages” (not “status”)?

Line 114: It was corrected to “stage”

Line 127: why you talk about TYLC in this section?

Thank you for this comment; we moved it to TYLC section “Tomato seedlings of TYLC tolerant ‘Charger’ or susceptible ‘FL-47’ cultivars were planted in spring 2019”

Line: 129-131: Consider adding the same information for the TYLC (section above).

It was added

Line 142: status or stages?

It was corrected to “Stages”

Line 144: add space between “23” and “mm”.

Line 177: It was added

Line 159: Consider to change this sub-title to: “UAV based data collection” to match with the previous sub-title “Laboratory data collection”.

Line 193: It was changed to “2.4 UAV based data collection”

Line 168: please check again this sentence.
It was checked

Line 190: consider to delete “the” before “specific”.

It was deleted “the” and add “a specific”.

Line 216: “when the M statistic value increases, it means more and better overlap separation”. Is that correct?

We corrected it to: “wider histograms (larger σ) will cause more overlap and less separation than narrow histograms (smaller σ) (Kaufman & Remer, 1994)”

Line 228: consider to add “Foe example” before “If the F…”.

It was corrected. Thanks

Line 240-244: Does these sentences belong to M&M (maybe the “2.1 Tomato yellow leaf curl (TYLC) sample collection” section)?

We tried to give a brief explanation of the inoculation of TYLC.

Line 290: Do you mean “Figure 4a”?

It was corrected to “Figure 4a”.

Line 319: cite the reference here based on journal’s instructions.

It was corrected to Abdulridha et al [4]

Table 2: consider to better describe columns 2 and 4. Maybe “STPD classification results (%)”.

It was modified as suggested.

Reviewer 2 Report

Review of remotesensing-887176 manuscrip
Abdulridha J et al. 2020. Laboratory and UAV-based identification and classification of Tomato yellow leaf curl, bacterial spot, and target spot diseases in tomato utilizing hyperspectral imaging and machine learning

This manuscript evaluate the potential of hyperspectral imaging both in the laboratory and in the field, for the early detection of diseases in tomatoes.

1. I was not able to find a definition of "healthy plant" or "healthy foliage". How did you get these?
2. How did you determine that a leaf was both "infected" and "asymptomatic"?
3. From the two comments above, I feel there is some degree of fuzziness in the definition of the 5 classes used for presenting and analysing the data (Figure 3 for example). This should be clarify with formal definitions of classes in all cases.
4. My understanding is that both TS and BS infected plants were from a TYLC tolerant and a TYLC susceptible crops. One would expect to see at least the following classes in Figure 4 "Healthy TYLC susceptible", "Healthy TYLC tolerant", "BS on TYLC susceptible", "BS on TYLC tolerant", "TS on TYLC susceptible", "TS on TYLC tolerant". This is not the case.
5. Lines 153-154: how did you avoid experimenter bias in selecting ROIs.
6. Line 173: I understand that six ROIs were selected from a random selection of ROIs. Again, how did you avoid experimenter bias.
7. Equation for the VIs: explain the notation which might not be obvious for someone new to this field. Make sure you use consistent notation. For example: why use R970 but NIR850. Should be NIR970. You must indicate that the numbers are the wavelength in nm... Check all equations for accuracy. For example, the notation for the RDVI is confusing as one might guess that "SQ" are constants (use a square root symbol (√) instead).
8. M statistic: I understand the idea behind this indicator: scale the difference in mean values by a measure of the dispersion in the data. However this raises many questions: why use the sum of the standard deviation instead of the geometric mean or a pooled standard deviation; what is the statistical distribution of this indicator (this is not far from the inverse of a coefficient of variation for which there are specific statistical distributions and tests). Is a Tukey's HSD test suitable for the M statistic? A more standard approach would be to perform an analysis of variance to test if the infection status (uncontrolled variable) and the crop variety (controlled variable) have a significant effect on a VI followed by a proper multiple comparisons of means when significant effects are detected with the analysis of variance. This approach avoids the definition of an indicator (M statistics) for which the appropriate statistical procedures are not defined.
9. Line 228: Mahalanobis.
10. Spectra in Figures 3 and 4: I guess that each curve represents the mean spectrum for the class. Please specify. If this is the case, it would be helpful to show the 95% confidence interval (or ± 1 standard deviation) around these mean spectra (transparent shade with the class color).
11. Laboratory vs field data: there are discrepencies between spectra for the same class acquired under laboratory and field conditions. This needs to be discussed. For example, in Figure 3b, there is a difference of about 0.1 in reflectance in the NIR region between the "Healthy" and the "TYLC susceptible" class while this difference is visibly smaller on Figure 3a. Also the "Healthy" spectrum is not parallel to the others in Figure 3a while it is on Figure 3b. This should be discussed. One possible explanation is that in the lab, you were looking at leaves lying roughly flat on a surface while in the field you were looking at the canopy reflectance. Discuss the difference between canopy reflectance and leaf reflectance.
12. Line 281 to 283: Again, without a definition of a "Healthy" plant, I am not sure of what this means.
13. Figure 4: Huge differences between reflectance recorded in the lab and in the field! This casts doubts about the proper handling of the leaves collected in the field and taken to the lab for measurements. For example, for the field data, the water absorption band around 970 nm is clearly visible while it seems to be greatly attenuated in the case of the lab data set.
14. In the scientific literature, concerns were expressed for the use of stepwise discriminant analysis for variable selection. In spectroscopy, more appropriate methods that handles correlation between independent variables are preferred. One such method is the PLSDA (Partial-Least-Square Discriminant Analysis). Proper use of PLSDA allows for the identification of discriminant spectral wavebands which is very useful in assessing and discussing the results. Please redo the discriminant analysis with PLSDA or other methods more suited to spectral data (SIMCA, SVM, ...).
15. Why perform pairwise discriminant analysis? Would be far superior to handle as a multiclass problem.
16. Not clear what Figures 5 and 6 shows. Columns are mean values? Intervals are confidence intervals at the 95% confidence level??? These figures are quite busy, consider splitting in 2 or more panels.
17. This whole section on selecting VIs based on Tukey's HSB on the M statistics is questionable for reasons outlined in point 8 above. I did not review in detail as I recommend that a more standard and suitable approach is used to analyse this data.

Overall, I do not recommend publication in its current version. More importantly:
a. Definition of the various classes should be clarify substantially (points 1 to 3 above).
b. Appropriate statistical procedures should be used in place of STDA on spectra and Tukey's HSD on M-estimator.
c. The differences observed between lab-based and field-based measurements should be highlighted and discussed.

Author Response

Reviewer 2

1. I was not able to find a definition of "healthy plant" or "healthy foliage". How did you get these?

New paragraph was added in the Material and Methods  line 117-122: “For each experiment, plants were physically separated in the field and plants were inoculated with either TS or BS. Plants were naturally infected with TYLC and Plant pathologist confirmed the pathogen associated with the foliar symptoms as either bacterial spot or the fungal pathogen of target spot. The healthy plants were not inoculated and grown in a separate field away from the infected plants and it was confirmed by the experts that they are not infected with any disease. Similarly, separate fields were selected for each case study (TYLC, BS and TS infected plants).

2. How did you determine that a leaf was both "infected" and "asymptomatic"?

If a plant is infected without visual symptoms, we consider it asymptomatic. New sections were added to explain the inoculation methods of TYLC, BS and TS; all diseased plants were under control and monitored by experts (pathologist and entomologist), who determined the disease development stages as well.

“2.2.1 Inoculation methods

2.2.1.1 Tomato yellow leaf curl disease

Naturally occurring populations of the viruliferous whitefly vector were sufficient to infect plants which was apparent from the symptoms on the plants.

2.2.1.2 Target spot

 Plants in plots were inoculated with Corynespora cassiicola on 15 Oct 18. Cultures of CC #19 and CC# 20 (kindly provided by Gary Vallad) were grown for 14 days on ¼ Potato dextrose agar + rifampicin and ampicillin (MilliporeSigma, St. Louis, MO 63103). Plates were flooded with sterile water and fungi was scraped from surface. The suspension containing mycelium and spores was filtered through three layers of cheese cloth and adjusted to approximately 104 spores’ ml-1 in sterile water. Inoculum was applied with a hand pump sprayer and plants were sprayed to run-off.  Confirmation of TS lesions was through microscopic examination of the lesions for presence of spores. No other disease was identified on these plants.

2.2.1.3 Bacterial spot

Plants were inoculated with Xanthomonas perforans races 3 and 4 on 17 Oct 18. Bacteria were grown in 25 ml of Difco Nutrient Broth (NB) overnight on a shaker incubator and transferred to 500 ml NB and incubated as before for 24 hours. Bacterial suspension was adjusted to 106 CFU ml-1 and applied to tomato plants to run-off using a hand pump sprayer. Lesions of bacterial spot were confirmed by re-isolation of the bacteria onto NA.”

3. From the two comments above, I feel there is some degree of fuzziness in the definition of the 5 classes used for presenting and analysing the data (Figure 3 for example). This should be clarifying with formal definitions of classes in all cases.
We hope the previous comments in point 1&2 would help for better understanding about how we identify each category.

4. My understanding is that both TS and BS infected plants were from a TYLC tolerant and a TYLC susceptible crops. One would expect to see at least the following classes in Figure 4 "Healthy TYLC susceptible", "Healthy TYLC tolerant", "BS on TYLC susceptible", "BS on TYLC tolerant", "TS on TYLC susceptible", "TS on TYLC tolerant". This is not the case.

The tolerant and susceptible varieties were only for the TYLC disease. There was no interaction between BS, TS, and H (separate fields were used for each experiment). They are all planted in same area (Southwest Florida Research and Education Center) but in separate fields to eliminate any interaction between the diseases. 

5. Lines 153-154: how did you avoid experimenter bias in selecting ROIs.

We tried to measure the spectral reflectance in different spots (infected and non-infected) to represent the whole leaf area.

6. Line 173: I understand that six ROIs were selected from a random selection of ROIs. Again, how did you avoid experimenter bias.

We changed the term “randomly” to “arbitrary” to better describe the selection method.

7. Equation for the VIs: explain the notation which might not be obvious for someone new to this field. Make sure you use consistent notation. For example: why use R970 but NIR850. Should be NIR970. You must indicate that the numbers are the wavelength in nm... Check all equations for accuracy. For example, the notation for the RDVI is confusing as one might guess that "SQ" are constants (use a square root symbol (√) instead).

All equations were checked and corrected; “nm” was added to each wavelength. The RDVI was corrected.

8. M statistic: I understand the idea behind this indicator: scale the difference in mean values by a measure of the dispersion in the data. However this raises many questions:

why use the sum of the standard deviation instead of the geometric mean or a pooled standard deviation;

 what is the statistical distribution of this indicator (this is not far from the inverse of a coefficient of variation for which there are specific statistical distributions and tests)?

Is a Tukey's HSD test suitable for the M statistic?

 A more standard approach would be to perform an analysis of variance to test if the infection status (uncontrolled variable) and the crop variety (controlled variable) have a significant effect on a VI followed by a proper multiple comparisons of means when significant effects are detected with the analysis of variance. This approach avoids the definition of an indicator (M statistics) for which the appropriate statistical procedures are not defined.
We agree that there are several methods to analyze this type of data. However, we prefer to use the M value and the developed methodology to be consistent with a previous study (Abdulridha et al 2019) in order to compare the results.

The M statistic value expresses the difference in means of two class histograms normalized by the sum of their standard deviations (σ). The extent to which M-statistics differ will depend on the width of the evaluated histograms. For the same difference in means, wider histograms (larger σ) will cause more overlap and less separation than narrow histograms (smaller σ) (Kaufman & Remer, 1994). The M-statistic was used in other studies too. For example Smith et al. (2017) used the M value to establish the best vegetation indices for the discrimination, where M < 1.0 indicates poor separation, M > 1.0 indicates good separation, and better separation occurs for larger M values. We removed the Tukey's HSD on M-estimator. There are several studies that have used this method:

Li, Shuang.; Xie, Y.; Dai, H.; Song, L, M-Statistic for Kernel Change-Point Detection. Part of Advance Neural information processing system 28. NIPS 2015. http://papers.nips.cc/book/advances-in-neural-information-processing-systems-28-2015.

Joosse, P.; Goslings. J C.; Luitse, J, S, K.; Ponsen, K J., M-Statistic Study: Arguments for Regional Trauma Databases. The Journal of Trauma. 2005, 58 (6), 1272.

Tebaldl, P.; Bonettim, M, Pagano, M.; M statistic commands: Interpoint distance distribution analysis. SASGE  Journal. 2011, 11 (2).

Kaufman, Y, J.; Remer, L, A.; Detection of forests using mid-IR reflectance: An application for aerosol studies. IEEE Transactions on Geoscience and Remote Sensing, 1994, 32, pp. 672-683.

Smith, A, M, S.; Dark, N, A.; Wooster, M, J.; Hudak, A, A.; Holden, Z, A.; Gibbsons, C, J.; Production of Landsat ETM + reference imagery of burned areas within Southern African savannahs: Comparison of methods and application to MODIS. International Journal of Remote Sensing, 2007 28, pp. 2753-2775.

9. Line 228: Mahalanobis.

Thanks, it was corrected

10. Spectra in Figures 3 and 4: I guess that each curve represents the mean spectrum for the class. Please specify. If this is the case, it would be helpful to show the 95% confidence interval (or ± 1 standard deviation) around these mean spectra (transparent shade with the class color).

We did not do it because the figures included several categories and doing this will be confusing to the reader, and there a lot of interactions between the wavebands in visible and NIR.

11. Laboratory vs field data: there are discrepancies between spectra for the same class acquired under laboratory and field conditions. This needs to be discussed. For example, in Figure 3b, there is a difference of about 0.1 in reflectance in the NIR region between the "Healthy" and the "TYLC susceptible" class while this difference is visibly smaller on Figure 3a. Also the "Healthy" spectrum is not parallel to the others in Figure 3a while it is on Figure 3b. This should be discussed. One possible explanation is that in the lab, you were looking at leaves lying roughly flat on a surface while in the field you were looking at the canopy reflectance. Discuss the difference between canopy reflectance and leaf reflectance.

New paragraph was added in discussion section to address this comment:  “The spectral reflectance showed differences between laboratory and field condition which might refer to the measurement conditions. In the laboratory, the leaf samples were laid down on one flat side and facing up the halogen light source with optimal temperature and humidity. Therefore, the reflectance of the light obtained from the leaf samples represent the object in ideal condition. The spectral reflectance in the field has different reflectance than the laboratory because of different sunlight angle(s) and light density, the angle of field of view, and weather conditions (e.g., clouds). Furthermore, the canopy structure of the plants included a different type of leaves of which some of them were infected and others were not or has a lower infection. In general, there was no proper way to select only symptomatic or asymptomatic leaves from images taken at 30 m height (UAV-based method). From the UAV taken images, it was difficult to determine the disease development stages, because in most cases plants included leaves with minor disease development stages and late symptoms.”

12. Line 281 to 283: Again, without a definition of a "Healthy" plant, I am not sure of what this means.

We have clarified it in previous comments (first two responses). Healthy means the plant is not under any kind of diseases or any stress factors.

13. Figure 4: Huge differences between reflectance recorded in the lab and in the field! This casts doubts about the proper handling of the leaves collected in the field and taken to the lab for measurements. For example, for the field data, the water absorption band around 970 nm is clearly visible while it seems to be greatly attenuated in the case of the lab data set.

As we discussed in the point 11, lab and field condition have different spectral reflectance in some cases. We just want to confirm all leaves were collected and measured in optimal conditions, so any variances in the spectral reflectance generated are based on plant condition and light condition.

14. In the scientific literature, concerns were expressed for the use of stepwise discriminant analysis for variable selection. In spectroscopy, more appropriate methods that handles correlation between independent variables are preferred. One such method is the PLSDA (Partial-Least-Square Discriminant Analysis). Proper use of PLSDA allows for the identification of discriminant spectral wavebands which is very useful in assessing and discussing the results. Please redo the discriminant analysis with PLSDA or other methods more suited to spectral data (SIMCA, SVM, ...).

We still want to use the STD method since we intend to compare our previous results with this new study. To better evaluate the results and the STD method, more parameters were added such as Wilks lambda, Chi square, and cross validation value. See Table 2. Additionally, beside the STD method we applied the radial basis function (RBF) to make sure our results are qualified with more than one classification method.

15. Why perform pairwise discriminant analysis? Would be far superior to handle as a multiclass problem.

Please see response to comments above.

16. Not clear what Figures 5 and 6 shows. Columns are mean values? Intervals are confidence intervals at the 95% confidence level??? These figures are quite busy, consider splitting in 2 or more panels.

The y-axis presents the M statistic values (with the error bar) for each VI (x-axis). We split this Figure to two VI groups (2 sub-figures).

17. This whole section on selecting VIs based on Tukey's HSB on the M statistics is questionable for reasons outlined in point 8 above. I did not review in detail as I recommend that a more standard and suitable approach is used to analyses this data.

Please see above comments, we deleted Tukey’s HSB from the task.

Round 2

Reviewer 2 Report

I think the revised manuscript adressed most of the concerns I expressed in my first review.